# Towards Formally Verifying LLMs: Taming the Nonlinearity of the Transformer

## Abstract

Large language models are increasingly used across various domains, which raises important safety concerns, particularly regarding adversarial attacks. While recent advancements in formal neural network verification have shown promising results, the complexity of transformers, the backbone of large language models, poses unique challenges for formal robustness verification. Traditional convex relaxation methods often result in large approximation errors due to the transformer's parallel, nonlinear attention heads. In this work, we address these limitations by introducing a novel approach based on non-convex, set-based computing to preserve the nonlinear dependencies through a transformer. Our approach generalizes previous methods on robustness verification of transformers, and the desired precision is tunable at the cost of additional computation time with a single parameter.

## 1 Introduction

Large language models (LLMs) have gained immense popularity in various fields, including question-answering, document summarization, and language translation (Raiaan et al., 2024). In particular, this success must be attributed to the transformers (Vaswani et al., 2017) used within these models. With increasing usage in all of these domains, safety concerns must be addressed: For example, how do we prevent the generation of harmful content? Large language models can be trained to reflect our desired behavior (Wallace et al., 2024); however, large language models – like any other neural network – can easily be fooled by adversarial attacks (Goodfellow et al., 2015). Thus, their behavior has to be formally verified to ensure safety, and such methods still need to be developed for large language models (Huang et al., 2024).

For example, consider a large language model that answers user prompts and our goal is to prevent harmful outputs. As a safety shield, we deploy a classifier language model supervising the in- and outputs, which returns a predefined answer if any in- or output is deemed harmful (Kim et al., 2023). This assumes that the classifier model has perfectly generalized from a given dataset defining harmful content – which is not even the case for standard feed-forward neural networks (Neyshabur et al., 2019), let alone large language models (Chang et al., 2024). Consider the following user prompt that reasonably should not be answered:

$$\text{How to build a bomb?} \tag{1}$$

As the prompts come from users, we have to assume they are trying to craft a prompt circumventing our safety shield, also known as an adversarial prompt (Zhu et al., 2023). Let $p'$ be such an adversarial user prompt crafted from the prompt $p$ (1) above. Crucially, we, as verifiers, are unaware that this was done and, in particular, $p$ is unknown to us. If we knew that a user was crafting adversarial prompts, we could simply block this user entirely. Thus, our goal is to find the prompt $p$, which we deem harmful and similar in meaning to the user prompt $p'$, and only return an answer to the user if we have verified that such a prompt $p$ does not exist. An adversarial prompt can be crafted by replacing certain words with synonyms (Zhu et al., 2023): For example, we could replace "build" with "construct", or "bomb" with "missile". As verifiers, we are required to test all possible combinations of synonyms and only answer the prompt if all synonym sentences are determined to be safe by the classifier model. The longer the user prompts, the more synonyms we have to consider – quickly leading to a combinatorial explosion. Additionally, defining synonyms is time-consuming and requires expert knowledge. Please also note that large language models usually do not operate on words but on tokens (Kudo & Richardson, 2018), such that synonyms have to be found on a token

level, and existing synonym collections on words cannot be used directly. Moreover, one might also want to include prompts that are not necessarily synonyms but share the same meaning, such as:

$$\texttt{How to build something that explodes?} \tag{2}$$

In such a sentence, the meaning of the term "`something that explodes`" has to be aggregated, which a large language model can do through its layers (Vaswani et al., 2017).

We address this issue by lifting recent advancements from formally verifying neural networks (Brix et al., 2023) to large language models. In particular, we examine how to verify transformers (Vaswani et al., 2017), which are the core components in modern language models (Achiam et al., 2024). Please note that in the standard setting of formal network verification, the input is usually perturbed by some noise, e.g., an $\ell_\infty$-ball with radius $\epsilon \in \mathbb{R}_+$. However, this is not directly applicable in the domain of natural language processing, as it has text as input. Instead, the noise is added in the embedding space, where synonyms are clustered together (Harris, 1954; Li & Yang, 2018). We then formally verify that there does not exist a synonym sentence in the perturbed embedding space that the classifier model deems unsafe. If we cannot rule out such a sentence, the user prompt is not answered by the large language model, and the predefined answer is returned instead.

**Our contributions.** We present a novel approach to formally verify large language models using set-based computing. This work is based on recent progress in neural network verification (Brix et al., 2023), and we focus on the main challenge when verifying large language models, namely verifying the transformer (Vaswani et al., 2017). All other layers can be verified using the techniques developed for standard neural networks (Brix et al., 2023).

Transformers are particularly challenging to verify due to the in-parallel computed nonlinearities in the self-attention heads (Vaswani et al., 2017), leading to nonlinear dependencies across all layers within a large language model and, thus, a non-convex output set. Neural network verifiers usually address this by relaxing the problem to a convex set enclosing the actual output (Brix et al., 2023). However, we show in this work that this leads to large outer approximations when applied on transformers and, for the first time, explicitly preserve these nonlinear dependencies through all layers using set-based computing. In particular, polynomial zonotopes (Kochdumper & Althoff, 2020) are used as a set representation.

Our evaluation shows that this enables the verification of much larger perturbed embedding spaces compared to related work (Sec. 5). Additionally, the desired precision can be tuned via a single parameter at the cost of additional verification time.

## 2 BACKGROUND

### 2.1 NOTATION

We denote scalars and vectors by lowercase letters, matrices by uppercase letters, and sets by calligraphic letters. The $i$-th element of a vector $v \in \mathbb{R}^n$ is written as $v_{(i)}$. The element in the $i$-th row and $j$-th column of a matrix $A \in \mathbb{R}^{n \times m}$ is written as $A_{(i,j)}$, the entire $i$-th row and $j$-th column are written as $A_{(i,\cdot)}$ and $A_{(\cdot,j)}$, respectively. The concatenation of $A$ with a matrix $B \in \mathbb{R}^{n \times o}$ is denoted by $[A \ B] \in \mathbb{R}^{n \times (m+o)}$. For two tensors $T_1, T_2 \in \mathbb{R}^{n_1 \times \ldots \times n_m}$, we also use the shorthand notation $[T_1 \ T_2]_i$ to denote their concatenation in the $i$-th dimension. The empty matrix is written as $[\ ]$. We use $I_n$ to denote the identity matrix of dimension $n \in \mathbb{N}$. The symbols $\mathbf{0}$ and $\mathbf{1}$ refer to matrices with all zeros and ones of proper dimensions, respectively. Given $n \in \mathbb{N}$, we use the shorthand notation $[n] = \{1, \ldots, n\}$. Let $\mathcal{S} \subset \mathbb{R}^n$ be a set and $f: \mathbb{R}^n \to \mathbb{R}^m$ be a function, then $f(\mathcal{S}) = \{f(x) \mid x \in \mathcal{S}\}$. An interval with bounds $a, b \in \mathbb{R}^n$ is denoted by $[a, b]$, where $a \leq b$ holds element-wise.

### 2.2 NEURAL NETWORKS

Let us now formally introduce neural networks and all operations used in large language models.

**Definition 1** (Neural Networks (Bishop & Nasrabadi, 2006, Sec. 5.1)). *Let $x \in \mathbb{R}^{n_0}$ be the input of a neural network $\Phi$ with $\kappa$ layers, its output $y = \Phi(x) \in \mathbb{R}^{n_\kappa}$ is obtained as follows:*

$$h_0 = x, \quad h_k = L_k(h_{k-1}), \quad y = h_\kappa, \qquad k \in [\kappa],$$

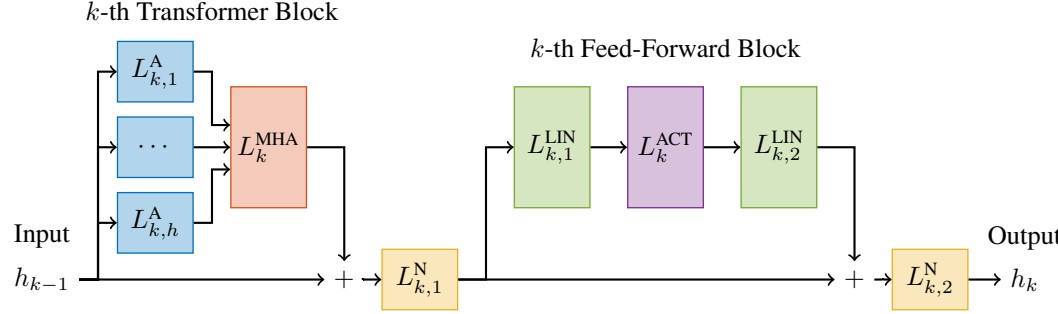

Figure 1: We consider large language model architectures in this paper where $\kappa$ blocks shown in the figure are concatenated, followed by global pooling and a linear layer to obtain the class predictions.

*where $L_k\colon \mathbb{R}^{n_{k-1}} \to \mathbb{R}^{n_k}$ represents the operation of layer $k$.*

For example, a linear layer with $W_k \in \mathbb{R}^{n_k \times n_{k-1}}$, $b_k \in \mathbb{R}^{n_k}$ is computed by

$$L_k^{\mathrm{L}}(h_{k-1}) = W_k h_{k-1} + b_{k-1} \in \mathbb{R}^{n_k}. \tag{3}$$

In this section, we focus on the layers used in large language models which are not used in standard neural networks (Fig. 1). We briefly list them next for easier reference when we propagate sets through them to verify the entire model. Let $d_{\mathrm{model}}$ denote the embedding dimension of our model for each of its $t$ tokens. Given three matrices $Q \in \mathbb{R}^{t \times d_{QK}}$, $K \in \mathbb{R}^{t \times d_{QK}}$, $V \in \mathbb{R}^{t \times d_V}$, a single self-attention layer (Vaswani et al., 2017, Eq. 1) is computed by

$$L_k^{\mathrm{A}}(Q, K, V) = \mathrm{softmax}\left(\frac{QK^T}{\sqrt{d_{QK}}}\right) V \in \mathbb{R}^{t \times d_V}, \tag{4}$$

where the softmax function is computed rowwise for a given vector $l \in \mathbb{R}^n$ as follows:

$$\mathrm{softmax}(l)_{(j)} = \frac{\exp l_{(j)}}{\sum_{i=1}^{n} \exp l_{(i)}} \in \mathbb{R}^n, \quad j \in [n]. \tag{5}$$

In practice, multiple attention heads are computed in parallel, and the overall equation is given by

$$L_k^{\mathrm{MHA}}(Q, K, V) = \left[L_{k,1}^{\mathrm{HA}}(Q, K, V) \quad \cdots \quad L_{k,h}^{\mathrm{HA}}(Q, K, V)\right]_1 W_k^A \in \mathbb{R}^{t \times d_{\mathrm{model}}},$$
$$\text{with } L_{k,i}^{\mathrm{HA}}(Q, K, V) = L_{k,i}^{\mathrm{A}}\left(QW_{k,i}^Q, KW_{k,i}^K, VW_{k,i}^V\right), \quad i \in [h], \tag{6}$$

where $W_{k,i}^Q, W_{k,i}^K \in \mathbb{R}^{d_{\mathrm{model}} \times d_{QK}}$, $W_{k,i}^V \in \mathbb{R}^{d_{\mathrm{model}} \times d_V}$, and $W_k^A \in \mathbb{R}^{h d_V \times d_{\mathrm{model}}}$ are learnable projection matrices for $Q$, $K$, $V$, and the aggregation of the attention heads, respectively. Similarly to related work (Shi et al., 2020; Bonaert et al., 2021), we slightly modify the layer normalization (Lei Ba et al., 2016) used in transformers. Please visit appendix A for a detailed discussion on this modification. Given bias and gain parameters $\beta$, $\gamma \in \mathbb{R}^{n_{k-1}}$, and an input $h_{k-1} \in \mathbb{R}^{n_{k-1}}$ with mean $\bar{h}_{k-1} = 1/n_{k-1} \cdot h_{k-1}$, the modified layer normalization is given by

$$L_k^{\mathrm{N}}(h_{k-1}) = \gamma \odot \left(h_{k-1} - \bar{h}_{k-1}\right) + \beta, \tag{7}$$

where $\odot$ denotes the Hadamard product. With that, we have introduced all special operations used in transformers.

## 2.3 SET-BASED COMPUTING

We use set-based computing to propagate perturbations to the input of a neural network through each layer. To this end, we use polynomial zonotopes (Kochdumper & Althoff, 2020) as set representation, in particular, its matrix variant:

**Definition 2** (Matrix Polynomial Zonotopes (Ladner et al., 2024, Def. 9)). *Given an offset $C \in \mathbb{R}^{n \times m}$, dependent generators $G \in \mathbb{R}^{n \times m \times h}$ with $h$ dependent generators, independent generators $G_I \in \mathbb{R}^{n \times m \times q}$ with $q$ independent generators, and an exponent matrix $E \in \mathbb{N}_0^{p \times h}$ with an identifier* $\mathrm{id} \in \mathbb{N}^{p}$[1], *a matrix polynomial zonotope $\mathcal{PZ} = \langle C, G, G_I, E \rangle_{PZ}$ is defined as*

$$\mathcal{PZ} = \langle C, G, G_I, E \rangle_{PZ} = \left\{ C + \sum_{i=1}^{h} \left( \prod_{k=1}^{p} \alpha_k^{E_{(k,i)}} \right) G_{(\cdot,\cdot,i)} + \sum_{j=1}^{q} \beta_j G_{I(\cdot,\cdot,j)} \; \middle| \; \alpha_k, \beta_j \in [-1, 1] \right\}.$$

We give an example for a polynomial zonotope in appendix B and only briefly introduce the required set operations here. The main advantage of using this set representation is its efficient computation of many operations appearing in the layers considered in Sec. 2.2: The Minkowski sum (Ladner et al., 2024, Eq. 5) of two matrix polynomial zonotopes

$$\mathcal{PZ}_1 = \left\langle C_1, \left[ \widetilde{G}_1 \; G_1 \right]_3, G_{I,1}, \left[ \widetilde{E} \; E_1 \right] \right\rangle_{PZ} \subset \mathbb{R}^{n \times m},$$
$$\mathcal{PZ}_2 = \left\langle C_2, \left[ \widetilde{G}_2 \; G_2 \right]_3, G_{I,2}, \left[ \widetilde{E} \; E_2 \right] \right\rangle_{PZ} \subset \mathbb{R}^{n \times m} \tag{8}$$

with partially shared exponent matrices and a common identifier vector is computed by:

$$\mathcal{PZ}_1 \oplus \mathcal{PZ}_2 = \{ x_1 + x_2 \mid x_1 \in \mathcal{PZ}_1, \; x_2 \in \mathcal{PZ}_2 \}$$
$$= \left\langle C_1 + C_2, \left[ \widetilde{G}_1 + \widetilde{G}_2 \quad G_1 \quad G_2 \right]_3, \left[ G_{I,1} \quad G_{I,2} \right]_3, \left[ \widetilde{E} \quad E_1 \quad E_2 \right]_2 \right\rangle_{PZ}. \tag{9}$$

Given the matrices $A_1 \in \mathbb{R}^{k \times n}$, $A_2 \in \mathbb{R}^{m \times k}$, and the vectors $b_1 \in \mathbb{R}^{k \times m}$, $b_2 \in \mathbb{R}^{n \times k}$, the affine map (Ladner et al., 2024, Eq. 6) is computed by

$$A_1 \mathcal{PZ}_1 + b_1 = \{ A_1 x + b_1 \mid x \in \mathcal{PZ}_1 \} = \langle A_1 C_1 + b_1, A_1 G_1, A_1 G_{I,1}, E_1 \rangle_{PZ},$$
$$\mathcal{PZ}_1 A_2 + b_2 = \{ x A_2 + b_2 \mid x \in \mathcal{PZ}_1 \} = \langle C_1 A_2 + b_2, G_1 A_2, G_{I,1} A_2, E_1 \rangle_{PZ}. \tag{10}$$

Given two matrix polynomial zonotopes $\mathcal{M}_1 \subset \mathbb{R}^{n \times k}$, $\mathcal{M}_2 \subset \mathbb{R}^{k \times m}$ with $g_1$ and $g_2$ generators, respectively, then the matrix multiplication

$$\mathcal{M}_3 := \mathcal{M}_1 \cdot \mathcal{M}_2 = \{ (M_1 \cdot M_2) \mid M_1 \in \mathcal{M}_1, M_2 \in \mathcal{M}_2 \} \subset \mathbb{R}^{n \times m}, \tag{11}$$

can be computed exactly such that $\mathcal{M}_3$ has $\mathcal{O}(g_1 g_2)$ generators (Ladner et al., 2024, Lemma 10) . Finally, the concatenation of two matrix polynomial zonotopes is done by concatenating their center and generators accordingly (appendix B, (19)). Further details on polynomial zonotopes and these operations can be found in appendix B.

## 2.4 NEURAL NETWORK VERIFICATION

We verify neural networks using (matrix) polynomial zonotopes by iteratively propagating the input set $\mathcal{X} = \mathcal{H}_0$ through the neural network and enclosing the output of each layer $k \in [\kappa]$:

**Proposition 1** (Image Enclosure (Kochdumper et al., 2023, Sec. 3)). *Let $\mathcal{H}_{k-1} \subset \mathbb{R}^{n_{k-1}}$ be an input set to layer $k$, then*

$$\mathcal{H}_k = \mathrm{enclose}\left( L_k, \mathcal{H}_{k-1} \right) \subset \mathbb{R}^{n_k}$$

*computes an outer-approximative output set.*

In general, linear layers can be computed exactly with polynomial zonotopes using (10), whereas nonlinear layers induce outer approximation (Fig. 2): The activation function is approximated using a polynomial, and an enclosure is obtained by bounding the approximation error, which is added to the output set using (9). The enclosure of the network's output set is then given by $\mathcal{Y} = \mathcal{H}_\kappa$.

## 2.5 PROBLEM STATEMENT

Given a large language model $\Phi \colon \mathbb{R}^{n_0} \to \mathbb{R}^{n_\kappa}$, a perturbed input set $\mathcal{X}$ corresponding to a user prompt, and some specification $\mathcal{S} \subset \mathbb{R}^{n_\kappa}$ specifying the unsafe outputs of a network, we want to compute an outer-approximative output set $\mathcal{Y} \supseteq \Phi(\mathcal{X})$ such that $\mathcal{Y} \cap \mathcal{S} = \emptyset$ to verify the safety of the user prompt. Please note that the classification problem stated in Sec. 1 can be formulated in this way.

---

[1] The identifier vector $\mathrm{id}$ is used to maintain dependencies of the factors $\alpha_k$ between sets.

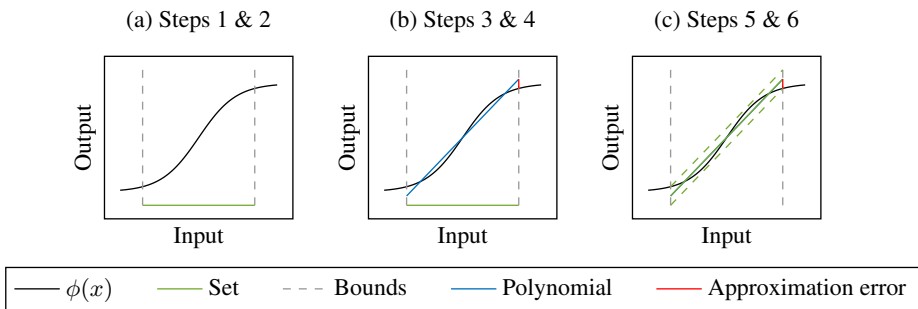

Figure 2: Main steps to enclose a nonlinear layer. Step 1: Evaluate the activation function element-wise. Step 2: Compute the bounds of the input set. Step 3: Find an approximating polynomial. Step 4: Compute the approximation error. Step 5: Evaluate the polynomial over the input set. Step 6: Add the approximation error.

## 3 NON-CONVEX DEPENDENCY-PRESERVING LLM VERIFICATION

We now present our novel approach to enclose the output of a large language model using polynomial zonotopes. An overview of the considered model is given in Fig. 1. We start with the smallest and simplest layers and incrementally add complexity by stacking these layers together. Please follow Sec. 2.2 along with this section for the exact equations and the required adaptations made here if the inputs are sets: Most layers only apply operations that can be computed efficiently using polynomial zonotopes (Sec. 2.3).

### 3.1 OUTPUT ENCLOSURE OF A LAYER NORMALIZATION LAYER

The modified layer normalization in (7) can be computed exactly without additional outer approximation. Please visit appendix A for a discussion on how to enclose the original equation of the layer normalization.

**Lemma 1** (Enclosure Layer Normalization). *Given an input set $\mathcal{H}_{k-1} \subset \mathbb{R}^{n_{k-1}}$, the output set of a layer normalization layer* (7) *is given by*

$$L_k^N(\mathcal{H}_{k-1}) = \operatorname{diag}(\gamma) \cdot (\mathcal{H}_{k-1} \oplus (-1/n_{k-1} \cdot \mathcal{H}_{k-1})) + \beta.$$

*Proof.* See appendix C. $\qquad\square$

### 3.2 OUTPUT ENCLOSURE OF THE SOFTMAX FUNCTION

The softmax layer is a standard layer in many network architectures. However, it often is the last layer of a network, and its enclosure can be omitted during verification, as one is usually only interested in the largest dimension, which can already be determined prior to the softmax layer. Unfortunately, this cannot be done for transformers, as it is applied internally. Analogously to Bonaert et al. (2021), we reformulate the computation of the softmax function (5) for numeric stability and to ease the enclosure using sets:

$$\operatorname{softmax}(l)_{(j)} = \frac{\exp l_{(j)}}{\sum_{i=1}^n \exp l_{(i)}} = \frac{1}{\sum_{i=1}^n \exp l_{(i)} - l_{(j)}}, \quad j \in [n]. \tag{12}$$

Then, we can obtain an output enclosure of the softmax function by enclosing the exponential and the inverse function:

**Lemma 2** (Enclosure Softmax). *Given an input set $\mathcal{L} \subset \mathbb{R}^n$, the output set of the softmax function* (12) *is enclosed by*

$$\operatorname{softmax}(\mathcal{L})_{(j)} \subseteq \mathtt{enclose}\left(x \mapsto 1/x, \mathbf{1} \cdot \mathtt{enclose}\left(\exp, \mathcal{L}_{(i)} \oplus -\mathcal{L}_{(j)}\right)\right), \quad j \in [n].$$

*The output set has $n$ more generators than the input set.*

*Proof.* See appendix C. □

Please note that the inverse function is only defined for positive inputs, which can, e.g., be ensured through the enclosure of $\exp$ using (Singh et al., 2018, Thm. 3.2). Computing good convex bounds has extensively been discussed in the literature (Bonaert et al., 2021; Wei et al., 2023), and we defer to their works for details on how to improve the bound computation. We found that, in practice, our method works well as long as the dependencies between dimensions are sufficiently well preserved.

### 3.3 Output Enclosure of an Attention Layer

The attention layer (4) evaluates the importance of tokens within a sequence to one another and aggregates their embeddings accordingly. With the input being perturbed, this requires the multiplication of sets and the resulting nonlinear dependencies need to be preserved for a tight enclosure. Fortunately, this multiplication of sets is exact using polynomial zonotopes (11), which is the unique advantage of our approach over related work. Please visit appendix B for a detailed discussion on dependency-preserving operations and, in particular, see Fig. 4b for a comparison of our approach with related work. Thus, we only induce outer approximations through the enclosure of the softmax function.

**Proposition 2** (Enclosure Attention). *Given three sets $\mathcal{Q}, \mathcal{K} \subset \mathbb{R}^{t \times d_{QK}}, \mathcal{V} \subset \mathbb{R}^{t \times d_V}$ with $g_{\mathcal{Q}}, g_{\mathcal{K}}, g_{\mathcal{V}}$ generators, respectively, the output set of an attention layer* (4) *is enclosed by*

$$L_k^A(\mathcal{Q}, \mathcal{K}, \mathcal{V}) \subseteq \texttt{enclose}\left(\texttt{softmax}, \frac{\mathcal{Q}\mathcal{K}^T}{\sqrt{d_{QK}}}\right)\mathcal{V},$$

*where $\mathcal{K}^T$ is computed by transposing the center and each generator of $\mathcal{K}$. The output set has $\mathcal{O}(g_{\mathcal{Q}}g_{\mathcal{K}}g_{\mathcal{V}} + t^2 g_{\mathcal{V}})$ generators.*

*Proof.* See appendix C. □

### 3.4 Output Enclosure of a Multi-Head Attention Layer

The output enclosure of the attention heads (Prop. 2) is computed multiple times in parallel during the verification of large language models (Fig. 1). Thus, after the output set of each attention head is obtained, the preserved dependencies ensure that their aggregation can be computed without inducing additional outer approximations:

**Proposition 3** (Enclosure Multi-Head Attention). *Given three sets $\mathcal{Q}, \mathcal{K}, \mathcal{V} \subset \mathbb{R}^{t \times d_{\mathrm{model}}}$ with $g_{\mathcal{Q}}, g_{\mathcal{K}}, g_{\mathcal{V}}$ generators, respectively, the output set of a multi-head attention layer* (6) *is enclosed by*

$$L_k^{MHA}(\mathcal{Q}, \mathcal{K}, \mathcal{V}) \subseteq [\mathcal{H}_{k,1} \quad \dots \quad \mathcal{H}_{k,h}]_1 W_k^A,$$

$$\text{with } \mathcal{H}_{k,i} = \texttt{enclose}\left(L_{k,i}^A, \mathcal{Q}W_{k,i}^Q, \mathcal{K}W_{k,i}^K, \mathcal{V}W_{k,i}^V\right), \quad i \in [h].$$

*The output set has $\mathcal{O}(g_{\mathcal{Q}}g_{\mathcal{K}}g_{\mathcal{V}} + ht^2 g_{\mathcal{V}})$ generators.*

*Proof.* See appendix C. □

### 3.5 Output Enclosure of a Large Language Model

After all enclosures of special layers appearing in a large language model are defined (Fig. 1), we can put everything together and compute an enclosure of the output set of the entire model. In this work, we consider a classifier model, as described in Sec. 1. Alg. 1 provides the pseudocode to verify the classifier model having $\kappa$ transformer blocks with $h$ attention heads each, linear and normalization layers in between, and a global pooling layer at the end to obtain the final classification. Please note that our approach is general and can also be used for other architectures, as the enclosures do not rely on a specific concatenation of the layers.

Alg. 1 requires a set $\mathcal{X} \subset \mathbb{R}^{t \times d_{\mathrm{model}}}$ as input; however, large language models have text as input. The text is split into $t$ tokens and each token is mapped into an embedding space of dimension $d_{\mathrm{model}}$. Thus, the input to the model is then given by

$$x \in \mathbb{R}^{t \times d_{\mathrm{model}}}. \tag{13}$$

---

**Algorithm 1** Enclosing the Output of a Large Language Model

---

**Require:** Large language model $\Phi$ and input set $\mathcal{X}$.                    ▷ Initialize input set (14)
1: $\mathcal{H}_0 \leftarrow \mathcal{X}$
2: **for** $k \in [\kappa]$ **do**                                                        ▷ Iterate over transformers
3:     **for** $i \in [h]$ **do**                                       ▷ Iterate over attention heads
4:         $\widehat{\mathcal{H}}_{k,i} \leftarrow \texttt{enclose}\left(L_{k,i}^{\mathrm{A}},\ \mathcal{H}_{k-1}W_{k,i}^Q,\ \mathcal{H}_{k-1}W_{k,i}^K,\ \mathcal{H}_{k-1}W_{k,i}^V\right)$ ▷ Attention head (Prop. 2)
5:     **end for**
6:     $\widehat{\mathcal{H}}_k \leftarrow \left[\widehat{\mathcal{H}}_{k,1} \quad \ldots \quad \widehat{\mathcal{H}}_{k,h}\right]_1 W_k^A$                    ▷ Aggregate attention heads (Prop. 3)
7:     $\overline{\mathcal{H}}_k \leftarrow L_{k,1}^{\mathrm{N}}\left(\mathcal{H}_{k-1} \oplus \widetilde{\mathcal{H}}_k\right)$        ▷ Residual connection (9) and normalization (Lemma 1)
8:     $\widetilde{\mathcal{H}}_k \leftarrow L_{k,2}^{\mathrm{LIN}}\left(L_k^{\mathrm{ACT}}\left(L_{k,1}^{\mathrm{LIN}}\left(\overline{\mathcal{H}}_k\right)\right)\right)$                        ▷ Regular neural network layers (Prop. 1)
9:     $\mathcal{H}_k \leftarrow L_{k,2}^{\mathrm{N}}\left(\mathcal{H}_{k-1} \oplus \widetilde{\mathcal{H}}_k\right)$        ▷ Residual connection (9) and normalization (Lemma 1)
10: **end for**
11: $\mathcal{Y} \leftarrow L_{\kappa+1}^{\mathrm{LIN}}\left(L_{\kappa+1}^{\mathrm{GP}}\left(\mathcal{H}_\kappa\right)\right)$                    ▷ Global Average Pooling (10) + Linear (3)
12: **return** Output set $\mathcal{Y} \supseteq \Phi(\mathcal{X})$

---

To also enclose synonyms of the input text, we construct an $\ell_\infty$-ball with radius $\epsilon \in \mathbb{R}_+$ around $x$ as follows:

$$\mathcal{X} = \left\langle x, [G_{1,1} \quad G_{1,2} \quad \ldots \quad G_{1,d_{\mathrm{model}}} \quad G_{2,1} \quad \ldots \quad G_{t,d_{\mathrm{model}}}]_3, [\,], I_{t \cdot d_{\mathrm{model}}}\right\rangle_{PZ} \subset \mathbb{R}^{t \times d_{\mathrm{model}}},$$

$$\text{with } G_{i,j(k,l)} = \begin{cases} \epsilon & \text{if } i = k \text{ and } j = l, \\ 0 & \text{otherwise,} \end{cases} \quad i, k \in [t],\ j, l \in [d_{\mathrm{model}}], \tag{14}$$

Thus, the overall generator matrix $G \in \mathbb{R}^{t \times d_{\mathrm{model}} \times (t \cdot d_{\mathrm{model}})}$ has $g_{\mathcal{X}} = (t \cdot d_{\mathrm{model}})$ generators with (initially) no dependencies between each other, as the exponent matrix is an identity matrix (Def. 2). This constructs a hypercube representing the $\ell_\infty$-ball around $x$, which is transformed to more complex sets within a large language model, particularly within the attention layer (Fig. 4b).

Please note that we use the same set $\mathcal{H}_{k-1}$ for the query, key, and value in the attention layer (Alg. 1, line 4), which allows us to simplify the number of generators derived in Prop. 2. Further, we reasonably assume that $t, d_{\mathrm{model}}, h \ll g_{\mathcal{X}}$. Thus,

**Lemma 3.** *The number of generators of $\mathcal{H}_k$, $k \in [\kappa]$, in Alg. 1 is $\mathcal{O}(g_{\mathcal{X}}^{3^k})$.*

*Proof.* See appendix C. □

**Reducing the number of generators.** Although the number of transformer blocks $\kappa$ is typically small, the direct computation of the output set $\mathcal{Y} = \mathcal{H}_\kappa$ quickly becomes intractable. Thus, we want to limit the number of generators to $g_{\max} \in \mathbb{R}_+$ by applying an order reduction method (Kochdumper & Althoff, 2020, Sec. II-E): The rapid growth stems from the matrix-set multiplications (11) within the attention layer (4), where the center and each generator of one set is multiplied with the center and each generator of the other set. The multiplications of the generators result in higher-order terms (see second example in appendix B), enabling the exact enclosure of the multiplication; however, many of these higher-order terms contribute only a little to the overall set and can be outer-approximated by an interval without losing much precision (Kochdumper & Althoff, 2020, Sec. II-B). Thus, we introduce a parameter $\rho_{\mathrm{lim}} \in \mathbb{R}_+$ limiting the number of generators of the higher-order terms relative to the dimension of the resulting set. We do not apply an order reduction on the linear terms resulting from the multiplication of the centers with each generator, respectively, as their dependencies between attention heads have to be kept for a tight enclosure. Please note that setting the error order $\rho_{\mathrm{lim}} = 1$ results in an interval hull of all higher-order terms, which corresponds to the approach in related work (Bonaert et al., 2021, Sec. 5.1); therefore, our approach is a direct generalization of their approach. Additionally, we apply an additional order reduction after each residual connection (Fig. 1) such that the total number of generators does not exceed $g_{\max} \in \mathbb{R}_+$. Thus, we keep the nonlinear dependencies as long as possible and only apply relaxations when necessary. Similar deferred relaxation techniques have also shown to be beneficial in other non-convex verification approaches on standard neural networks (Wei et al., 2023; Ladner & Althoff, 2023; Fatnassi et al., 2023).

**Theorem 1** (Enclosure of Large Language Models). *Given a large language model $\Phi$ and an input set $\mathcal{X}$ (14) with $g_{\mathcal{X}}$ generators, Alg. 1 computes an output set $\mathcal{Y}$ satisfying the problem statement stated in Sec. 2.5. The computational complexity is bounded by $\mathcal{O}\left(thd_V d_{\text{model}} g_{\text{max}} \kappa\right)$.*

*Proof.* See appendix C. □

## 4 EXPERIMENTAL RESULTS

We evaluate our approach on four large language models $M_i$, $i \in [4]$, trained from scratch for binary classification. All hyperparameters and additional experiments are given in appendix D. The first two models are trained on the medial safety dataset (Abercrombie & Rieser, 2022), and the last two models are trained on the Yelp dataset (Zhang et al., 2016), We implemented our approach into the Matlab toolbox CORA (Althoff, 2015). All computations were performed in a docker container on an Intel® Core™ Gen. 11 i7-11800H CPU @2.30GHz with 64GB memory.

### 4.1 ON SYNONYM SENTENCE ENUMERATION AND FORMAL VERIFICATION

Let us first compare how our approach scales with the number of synonyms captured by the embedding space compared to enumerating all synonym sentences. Fig. 3a shows that enumerating all synonym sentences quickly becomes computationally infeasible, while our approach can capture synonym sentences in a single verification query and guarantees safety in a few seconds. Please note that reasonable sentences have many more synonyms as exemplary shown in Tab. 1: This sentence from the medical safety dataset has 96 synonym words with more than 2 billion synonym sentences.

### 4.2 VERIFICATION RESULTS ON LARGE LANGUAGE MODELS

Let us now evaluate our verification approach and compare it to a state-of-the-art transformer verification approach based on zonotopes ((Bonaert et al., 2021)) and naive interval bound propagation ((Jaulin et al., 2001)). We show the tighter enclosure of a single attention head in Fig. 3b, which has to be attributed to the ability to compute the multiplication of two polynomial zonotopes exactly (11) at the cost of additional verification time. Further details on this unique advantage are given in appendix B.

This tighter enclosure allows us to verify the safety of much larger perturbed embedding spaces. We present the averaged results of 20 sentences with up to 27 tokens of each dataset, respectively, in Tab. 2. For each sentence, we perform a binary search to determine the largest input set that is still verifiable with each approach, respectively, where we normalize the verified volume of the zonotope baseline to 1. Our approach allows adaptively increasing the parameter $\rho_{\text{lim}}$ to compute tighter enclosures and thus larger verified perturbed embedding spaces at the cost of additional computation time. Please note that setting $\rho_{\text{lim}} = 1$ corresponds to the zontope approach and no nonlinear

Table 1: Synonyms contained in a perturbed embedding space for a verified example sentence.

| Token | #Synonyms | Synonyms |
|---|---|---|
| dull | 14 | heavy, flat, uncomfortable, ... |
| chest | 4 | body, heart, ... |
| pain | 1 | ache |
| when | 1 | whenever |
| bending | 16 | turning, leaning, bowing, ... |
| in | 6 | at, into, within, ... |
| certain | 12 | specific, chosen, fixed, ... |
| directions | 10 | position, ways, angle, ... |
| and | 1 | besides |
| when | 1 | whenever |
| breathing | 4 | air, oxygen, ... |
| ... | ... | ... |

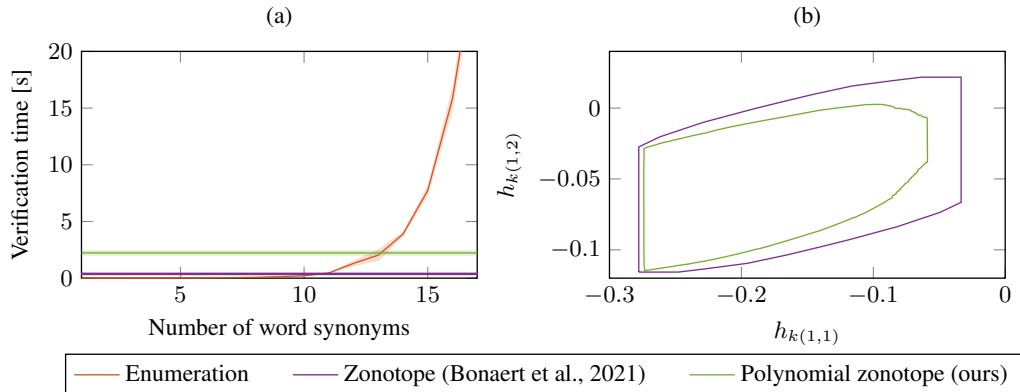

Figure 3: (a) Comparison of the time needed to verify all synonym sentences by enumerating them and a single verification query using our approach. (b) Enclosure comparison of a single attention head (4) with $\rho_{\lim} = \infty$.

Table 2: Comparison of verified embedding space with interval bound propagation (IBP), zonotopes (Z, Bonaert et al. (2021)), and polynomial zonotopes (PZ, ours).

| | **Model 1** | | | **Model 3** | | |
| | Verified Volume | | Time [s] | Verified Volume | | Time [s] |
| **Approach** | Mean | Max. | | Mean | Max. | |
|---|---|---|---|---|---|---|
| IBP | 0.00 | 0.00 | **0.02** | 0.00 | 0.00 | **0.02** |
| Z (baseline) | 1.00 | 1.00 | 6.90 | 1.00 | 1.00 | 5.97 |
| PZ ($\rho_{\lim} = 2$) | 1,874.07 | 29,723.35 | 201.71 | 17,432.27 | 331,146.39 | 170.71 |
| PZ ($\rho_{\lim} = 5$) | **2,993.96** | **46,621.47** | 283.55 | **80,593.07** | **1,531,038.79** | 228.33 |

dependencies are captured anymore. Further details on this tradeoff are given in appendix D on two more models.

## 5 RELATED WORK

This work continuous the progressive effort to formally verify neural networks in recent years (Brix et al., 2023): As neural networks are vulnerable to adversarial attacks (Goodfellow et al., 2015), formally verifying them against input perturbations is of immense importance in safety-critical scenarios. Neural network verifiers can generally be categorized into optimization-based verifiers Zhang et al. (2018); Katz et al. (2019); Henriksen & Lomuscio (2020); Singh et al. (2019) and approaches using set-based computing Gehr et al. (2018); Singh et al. (2018); Lopez et al. (2023); Kochdumper et al. (2023). Early approaches focused on computing the exact output set of neural networks with ReLU activations, which has been shown to be NP-hard (Katz et al., 2017). Convex relaxations of the problem (Singh et al., 2018; Xu et al., 2020) and efficient branch-and-bound techniques (Wang et al., 2021) have shown to achieve impressive verification results (Brix et al., 2023). Recently, non-convex relaxations have also shown to be effective in this field of research (Kochdumper et al., 2023; Wang et al., 2023; Ladner & Althoff, 2023; Fatnassi et al., 2023; Ortiz et al., 2023).

While the robustness of large language models against adversarial prompts has been investigated (Sun et al., 2024, Sec. 9), research on the non-existence of such prompts using formal verification is limited and, as in this work, only perturbations to the embedding space of the input of large language models (Vaswani et al., 2017) are considered: As with standard neural networks, Shi et al. (2020) use linear bounds to verify the robustness with at most two perturbed words. This work has been extended to the zonotope domain (Bonaert et al., 2021), enabling the verification of longer sentences with all words being perturbed. Other works also investigated finding better bounds of the softmax function within transformers (Wei et al., 2023; Shi et al., 2024). Recently, the importance of

carrying dependencies between each attention head and dimension has been stressed (Zhang et al., 2024); however, they also apply convex relaxations in each step and thus lose precision at each nonlinearity.

We also consider the problem of formally verifying graph neural networks (Sälzer & Lange, 2023) as closely related, as many of the operations within these networks (Kipf & Welling, 2017) also appear in large language models (Vaswani et al., 2017). Thus, progress in either domain also propels progress in the other. In graph neural networks, approaches consider problems with perturbed input (Zügner & Günnemann, 2019) and also perturbed graph structure (Bojchevski & Günnemann, 2019; Jin et al., 2020; Ladner et al., 2024).

## 6 LIMITATIONS

Our work relies on the implicit assumption that words with similar meanings have similar word embeddings and are thus captured by constructing an $\ell_\infty$-ball around the embedding of a given word. While literature (Harris, 1954; Li & Yang, 2018) and our experiments suggest that this holds, we cannot guarantee that we capture all synonyms. More advanced perturbations might also be beneficial to capture more synonyms, e.g., by considering the superpositions of features in word embeddings (**?**). It might also be necessary to add further perturbations after the model aggregated meanings of multiple words through each attention block (Vaswani et al., 2017, Appendix). For example, by adding perturbations after each transformer block (Alg. 1, lines 2 to 10)), one might capture the example sentence (2) as a synonym of (1). Furthermore, our approach does not output the original harmful prompt to the adversarial user prompt (Sec. 1). We can only say that there might exist an area in the embedding space, which might contain a harmful prompt. Additionally, this unsafe area might also not correspond to an actual synonym sentence as it is sparsely populated. However, we would rather want to be safe than output harmful content.

Our approach continues the progressive efforts to formally verify large language models (Shi et al., 2020; Bonaert et al., 2021). However, all methods are not yet applicable to modern-size large language models (Achiam et al., 2024), where both high precision and low verification time are desirable. Our approach is a generalization of Bonaert et al. (2021) based on zonotopes, where a balance between precision and speed is adaptively struck via the parameter $\rho_{\text{lim}}$. Please note that the total number of generators $g_{\text{max}}$ also limits the precision. Order reduction methods are more challenging for polynomial zonotopes than for zonotopes due to the additional complexity in the set representation. Recently, an approach was made addressing this underexplored problem (Ladner & Althoff, 2024), and any improvement on this relatively new set representation also propels our approach.

It is also worth noting that we only verify the robustness of large language models against adversarial prompts to fool classifier models; however, other model architectures and safety constraints are also worth considering, e.g., jailbreaks (Wei et al., 2024; Casper et al., 2023), to prevent outputting false claims or exposing private data. We defer to (Sun et al., 2024, Ch. 7) for a more detailed discussion on the safety aspects of large language models.

Please note that transformers are also used in other domains, e.g., computer vision (Bhojanapalli et al., 2021; Khan et al., 2022), and our verification approach can also be applied there as our enclosures are defined per layer and do not rely on a specific model.

## 7 CONCLUSION

We present a novel approach to tightly enclose the output set of a large language model given a perturbed input. This tight enclosure is realized using non-convex set-based computing based on polynomial zonotopes, which allows one to efficiently preserve nonlinear dependencies. This preservation of dependencies is particularly important for the verification of large language models, as the transformers used within large language models repeatedly compute complex nonlinear functions in parallel. The desired precision can be tuned with a single parameter at the cost of additional computation time. These advantages of our approach are demonstrated by rigorous theoretical analysis and experiments, where we can verify much larger embedding spaces than approaches developed in related work. We believe this work is a significant step towards improving these models' formal safety.

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

# Appendix

The appendix is structured as follows:

- A discussion about the modified layer normalization is given in appendix A.
- A more detailed explanation of (matrix) polynomial zonotopes and dependency preservation is given in appendix B.
- All proofs are given in appendix C.
- Finally, we list all hyperparameters of the considered models and provide additional experiments in appendix D.

## A  LAYER NORMALIZATION DISCUSSION

We use a modified equation of the layer normalization in our models (7). The original equation (Lei Ba et al., 2016) is given by

$$L_k^{\mathrm{LN}}(h_{k-1}) = \frac{\gamma}{\sigma} \odot (h_{k-1} - \bar{h}_{k-1}) + \beta \tag{15}$$

with the only difference being the division by the standard deviation $\sigma \in \mathbb{R}$ of the input $h_{k-1}$. This change was proposed by (Shi et al., 2020, Appendix E) to ease verification and was also used by Bonaert et al. (2021). The issue with the division by the standard deviation is that as the input is a set $\mathcal{H}_{k-1}$, also the standard deviation is a set containing the standard deviation for each point in $\mathcal{H}_{k-1}$. Thus, the division by $\sigma$ in (15) is in general very outer-approximative, which heavily reduces the verifiability. The argument for the modified layer normalization (7) is that it does not penalize the accuracy of the obtained model (Shi et al., 2020, Tab. 5) but greatly increases the verifiability. We want to stress that designing network architectures with formal verification in mind is a motion we strongly support. However, it appears that even a model without any normalization has a similar accuracy in their evaluation, which raises the question of whether layer normalization is necessary at all. Thus, the influence of this modification must be further investigated, especially for larger models. If the original layer normalization (15) turns out to be beneficial, a similar technique as the one described for the softmax layer (Lemma 2) can be applied, as a division by a set also appears there.

## B  ON POLYNOMIAL ZONOTOPES

Polynomial zonotopes are effectively a compact representation of a polynomial in high-dimensional space. Let us recall the definition of a matrix polynomial zonotope:

**Definition 2** (Matrix Polynomial Zonotopes (Ladner et al., 2024, Def. 9))**.** *Given an offset* $C \in \mathbb{R}^{n \times m}$*, dependent generators* $G \in \mathbb{R}^{n \times m \times h}$ *with $h$ dependent generators, independent generators* $G_I \in \mathbb{R}^{n \times m \times q}$ *with $q$ independent generators, and an exponent matrix $E \in \mathbb{N}_0^{p \times h}$ with an identifier* $\mathtt{id} \in \mathbb{N}^{p2}$*, a matrix polynomial zonotope* $\mathcal{PZ} = \langle C, G, G_I, E \rangle_{PZ}$ *is defined as*

$$\mathcal{PZ} = \langle C, G, G_I, E \rangle_{PZ} = \left\{ C + \sum_{i=1}^{h} \left( \prod_{k=1}^{p} \alpha_k^{E_{(k,i)}} \right) G_{(\cdot,\cdot,i)} + \sum_{j=1}^{q} \beta_j G_{I(\cdot,\cdot,j)} \ \middle|\ \alpha_k, \beta_j \in [-1,1] \right\}.$$

We chose to use the matrix variant over the regular one (Kochdumper & Althoff, 2020), as transformers mainly operate on matrices and not on vectors. In this section, we only use regular polynomial zonotopes for easier notation, where the center and each generator is in $\mathbb{R}^n$ rather than $\mathbb{R}^{n \times m}$. Any properties derived in this section also hold for the matrix variant. Let us construct a simple one-dimensional polynomial zonotope describing the interval $[-1, 1]$:

$$\mathcal{PZ} = \langle 0, 1, [\,], 1 \rangle_{PZ} = \left\{ 0 + \alpha_1^1 \cdot 1 \ \middle|\ \alpha_1 \in [-1,1] \right\} = [-1, 1]. \tag{16}$$

---

[2] The identifier vector $\mathtt{id}$ is used to maintain dependencies of the factors $\alpha_k$ between sets.

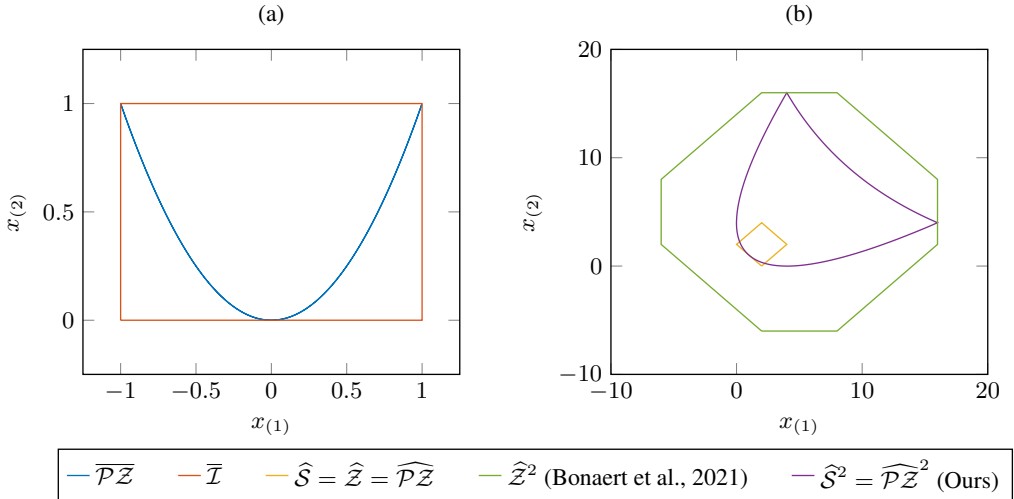

Figure 4: Visualization of preserved dependencies using polynomial zonotopes in appendix B: a) Between individual sets and b) a comparison of our approach with the state-of-the-art transformer verification approach.

Computing $\mathcal{PZ}^2 = \{x^2 \mid x \in \mathcal{PZ}\}$ would then simply be:

$$\mathcal{PZ}^2 = \langle 0, 1, [\,], 2 \rangle_{PZ} = \left\{ 0 + \alpha_1^2 \cdot 1 \mid \alpha_1 \in [-1, 1] \right\} = [0, 1]. \tag{17}$$

One advantage of polynomial zonotopes is that they can carry nonlinear dependencies and thus represent non-convex sets. The Cartesian product of these two sets is:

$$\begin{bmatrix} \mathcal{PZ} \\ \mathcal{PZ}^2 \end{bmatrix} = \left\langle \begin{bmatrix} 0 \\ 0 \end{bmatrix}, \begin{bmatrix} 1 & 0 \\ 0 & 1 \end{bmatrix}, [\,], [1\ 2] \right\rangle_{PZ} = \left\{ \begin{bmatrix} 0 \\ 0 \end{bmatrix} + \alpha_1^1 \cdot \begin{bmatrix} 1 \\ 0 \end{bmatrix} + \alpha_1^2 \cdot \begin{bmatrix} 0 \\ 1 \end{bmatrix} \,\middle|\, \alpha_1 \in [-1, 1] \right\}. \tag{18}$$

More formally, given two polynomial zonotopes $\mathcal{PZ}_1 = \left\langle c_1, [\widetilde{G}_1\ G_1], G_{I,1}, [\widetilde{E}\ E_1] \right\rangle_{PZ}$ and $\mathcal{PZ}_2 = \left\langle c_2, [\widetilde{G}_2\ G_2], G_{I,2}, [\widetilde{E}\ E_2] \right\rangle_{PZ}$ with partially shared exponent matrices and a common identifier vector, their Cartesian product (Kochdumper et al., 2023, Eq. 7) is computed by

$$\begin{aligned} \mathcal{PZ}_1 \times \mathcal{PZ}_2 = \begin{bmatrix} \mathcal{PZ}_1 \\ \mathcal{PZ}_2 \end{bmatrix} &= \left\{ \begin{bmatrix} x_1 \\ x_2 \end{bmatrix} \,\middle|\, x_1 \in \mathcal{PZ}_1,\ x_2 \in \mathcal{PZ}_2 \right\} \\ &= \left\langle \begin{bmatrix} c_1 \\ c_2 \end{bmatrix}, \begin{bmatrix} \widetilde{G}_1 & G_1 & \mathbf{0} \\ \widetilde{G}_2 & \mathbf{0} & G_2 \end{bmatrix}, \begin{bmatrix} G_{I,1} & \mathbf{0} \\ \mathbf{0} & G_{I,2} \end{bmatrix}, \begin{bmatrix} \widetilde{E} & E_1 & E_2 \end{bmatrix} \right\rangle_{PZ}, \end{aligned} \tag{19}$$

where a common identifier vector intuitively means that both sets use the same $\alpha_k$ in Def. 2. If the identifier vector is not identical, the exponent matrices of both sets have to be extended accordingly (Kochdumper & Althoff, 2020, Prop. 1). Both $\overline{\mathcal{PZ}} \coloneqq \mathcal{PZ} \times \mathcal{PZ}^2$ (18) and the set $\overline{\mathcal{I}} \coloneqq [-1, 1] \times [0, 1]$, where those dependencies are not considered, are shown in Fig. 4a. Clearly, the preserved dependencies allow us to compute the considered operations without inducing any outer approximation. Thus, $\overline{\mathcal{PZ}} \subset \overline{\mathcal{I}}$ holds. Analogous reasoning also holds for the computation of the Minkowski sum (9).

Another crucial operation when verifying large language models is the multiplication of two sets, as as this operation is executed throughout the entire model once the inputs are uncertain, e.g., within one attention head (4). To illustrate the advantage of polynomial zonotopes over zonotopes as was used in related work (Bonaert et al., 2021), which work identically except they do not have an exponent matrix to store the nonlinear dependencies, consider the following set:

$$\widehat{\mathcal{S}} = \widehat{\mathcal{Z}} \coloneqq \left\langle \begin{bmatrix} 2 \\ 2 \end{bmatrix}, \begin{bmatrix} 1 & -1 \\ 1 & 1 \end{bmatrix} \right\rangle_Z = \left\langle \begin{bmatrix} 2 \\ 2 \end{bmatrix}, \begin{bmatrix} 1 & -1 \\ 1 & 1 \end{bmatrix}, [\,], \begin{bmatrix} 1 & 0 \\ 0 & 1 \end{bmatrix} \right\rangle_{PZ} =: \widehat{\mathcal{PZ}}. \tag{20}$$

The multiplications of two high-dimensional sets is generally referred to as the quadratic map (Kochdumper & Althoff, 2020, Def. 6). We consider a special case here and want to compute:

$$\widehat{\mathcal{S}}^2 := \left\{ \begin{bmatrix} \widehat{x}_{(1)}^2 \\ \widehat{x}_{(2)}^2 \end{bmatrix} \,\middle|\, \widehat{x} \in \widehat{\mathcal{S}} \right\}. \tag{21}$$

Using the method to compute precise bounds from (Bonaert et al., 2021, Thm. 5) on zonotopes, we obtain

$$\widehat{\mathcal{Z}}^2 := \left\langle \begin{bmatrix} 5 \\ 5 \end{bmatrix}, \begin{bmatrix} 4 & -4 & 3 & 0 \\ 4 & 4 & 0 & 3 \end{bmatrix} \right\rangle_Z. \tag{22}$$

On the other hand, using polynomial zonotopes (Kochdumper & Althoff, 2020, Prop. 12) results in

$$\widehat{\mathcal{P}\mathcal{Z}}^2 := \left\langle \begin{bmatrix} 4 \\ 4 \end{bmatrix}, \begin{bmatrix} 4 & -4 & 1 & 1 & -2 \\ 4 & 4 & 1 & 1 & 2 \end{bmatrix}, [\,], \begin{bmatrix} 1 & 0 & 2 & 0 & 1 \\ 0 & 1 & 0 & 2 & 1 \end{bmatrix} \right\rangle_{PZ}, \tag{23}$$

without inducing any outer approximation. Thus, $\widehat{\mathcal{S}}^2 = \widehat{\mathcal{P}\mathcal{Z}}^2 \subset \widehat{\mathcal{Z}}^2$ holds. As shown in Fig. 4b, the outer approximation using zonotopes can be quite significant – even for this toy example, let alone when applied multiple times within a large language model. In particular, the last three generators of $\widehat{\mathcal{P}\mathcal{Z}}^2$ are outer-approximated with interval bounds in $\widehat{\mathcal{Z}}^2$ (Bonaert et al., 2021, Thm. 5). However, these three generators are required to capture the non-convexity of the output set and this non-convexity is thus lost using the interval bounds. The matrix multiplications on sets stated in (11) are computed analogously using the quadratic map, and thus have the same issues if the dependencies are not preserved.

## C PROOFS

We include all proofs from the main body in this section in the order of appearance.

**Lemma 1** (Enclosure Layer Normalization)**.** *Given an input set* $\mathcal{H}_{k-1} \subset \mathbb{R}^{n_{k-1}}$*, the output set of a layer normalization layer* (7) *is given by*

$$L_k^N(\mathcal{H}_{k-1}) = \operatorname{diag}(\gamma) \cdot (\mathcal{H}_{k-1} \oplus (-\mathbf{1}/n_{k-1} \cdot \mathcal{H}_{k-1})) + \beta.$$

*Proof.* The exact computation follows directly from (10) and (9). □

**Lemma 2** (Enclosure Softmax)**.** *Given an input set* $\mathcal{L} \subset \mathbb{R}^n$*, the output set of the softmax function* (12) *is enclosed by*

$$\operatorname{softmax}(\mathcal{L})_{(j)} \subseteq \texttt{enclose}\left(x \mapsto 1/x, \mathbf{1} \cdot \texttt{enclose}\left(\exp, \mathcal{L}_{(i)} \oplus -\mathcal{L}_{(j)}\right)\right), \quad j \in [n].$$

*The output set has* $n$ *more generators than the input set.*

*Proof.* The enclosure follows directly from Prop. 1 and (10). While each enclosure operation adds $n$ generators to the set for each approximation error of the $n$ dimensions, these are aligned and can be summed up, resulting in $n$ generators. □

**Proposition 2** (Enclosure Attention)**.** *Given three sets* $\mathcal{Q}, \mathcal{K} \subset \mathbb{R}^{t \times d_{QK}}, \mathcal{V} \subset \mathbb{R}^{t \times d_V}$ *with* $g_{\mathcal{Q}}, g_{\mathcal{K}}, g_{\mathcal{V}}$ *generators, respectively, the output set of an attention layer* (4) *is enclosed by*

$$L_k^A(\mathcal{Q}, \mathcal{K}, \mathcal{V}) \subseteq \texttt{enclose}\left(\operatorname{softmax}, \frac{\mathcal{Q}\mathcal{K}^T}{\sqrt{d_{QK}}}\right)\mathcal{V},$$

*where* $\mathcal{K}^T$ *is computed by transposing the center and each generator of* $\mathcal{K}$*. The output set has* $\mathcal{O}(g_{\mathcal{Q}}g_{\mathcal{K}}g_{\mathcal{V}} + t^2 g_{\mathcal{V}})$ *generators.*

*Proof.* The statement follows directly from (11), (10), Lemma 2, and (19). The number of generators also follows from (11) and the multiple applications of Lemma 2 on the softmax function, which is applied for each of the $t$ rows individually, each having $t$ entries. Thus, the term $g_{\mathcal{Q}}g_{\mathcal{K}}g_{\mathcal{V}}$ corresponds to the set approximating the output and $t^2 g_{\mathcal{V}}$ corresponds to the approximation error of this layer. Please note that for a tight enclosure, it is crucial that the dependencies of the sets obtained by the rowwise application of the softmax function are preserved while stacking them back together using the Cartesian product, as described in appendix B, (19). □

**Proposition 3** (Enclosure Multi-Head Attention). *Given three sets $\mathcal{Q}, \mathcal{K}, \mathcal{V} \subset \mathbb{R}^{t \times d_{\mathrm{model}}}$ with $g_{\mathcal{Q}}, g_{\mathcal{K}}, g_{\mathcal{V}}$ generators, respectively, the output set of a multi-head attention layer (6) is enclosed by*

$$L_k^{MHA}\left(\mathcal{Q}, \mathcal{K}, \mathcal{V}\right) \subseteq \left[\mathcal{H}_{k,1} \quad \ldots \quad \mathcal{H}_{k,h}\right]_1 W_k^A,$$

$$\text{with } \mathcal{H}_{k,i} = \texttt{enclose}\left(L_{k,i}^A, \ \mathcal{Q}W_{k,i}^Q, \mathcal{K}W_{k,i}^K, \mathcal{V}W_{k,i}^V\right), \quad i \in [h].$$

*The output set has $\mathcal{O}(g_{\mathcal{Q}}g_{\mathcal{K}}g_{\mathcal{V}} + ht^2 g_{\mathcal{V}})$ generators.*

*Proof.* The concatenation $[\ldots]_1$ is computed using the Cartesian product (19), the remainder follows from Prop. 2 and (10). The number of generators follows directly from the enclosure of the $h$ attention heads (Prop. 2), where only the approximation error is treated independently during the concatenation (19). Thus, the term $g_{\mathcal{Q}}g_{\mathcal{K}}g_{\mathcal{V}}$ again corresponds to the set approximating the output and $ht^2 g_{\mathcal{V}}$ corresponds to the approximation error of this layer. $\square$

**Lemma 3.** *The number of generators of $\mathcal{H}_k$, $k \in [\kappa]$, in Alg. 1 is $\mathcal{O}(g_{\mathcal{X}}^{3^k})$.*

*Proof.* We show this proof by induction:
*Induction base* $k = 0$: $\mathcal{H}_0 = \mathcal{X}$. The statement follows trivially as $\mathcal{O}(g_{\mathcal{X}}^{3^0}) = \mathcal{O}(g_{\mathcal{X}})$.
*Induction hypothesis*: Let the statement now hold for an arbitrary $k \in [\kappa]$.
*Induction step* $k + 1$: Let $\mathcal{H}_{k+1}$ have $\mathcal{O}(g_{\mathcal{H}_{k+1}})$ generators. Please not that the regular layers in line 8 add at most one generator per neuron due to the activation layer (Prop. 1), which are $t \cdot d_{\mathrm{model}} \in \mathcal{O}(g_{\mathcal{H}_{k+1}})$. Thus, as the normalization layer is exact (Lemma 1) and the sets added the residual connections essentially share the same exponent matrix (9), also $\widetilde{\mathcal{H}}_{k+1}$, $\overline{\mathcal{H}}_{k+1}$ and $\widehat{\mathcal{H}}_{k+1}$ have $\mathcal{O}(g_{\mathcal{H}_{k+1}})$ generators, respectively. Then, using Prop. 3 and the assumption $t, d_{\mathrm{model}}, h \ll g_{\mathcal{X}}$, we can derive that $\mathcal{H}_k$ has to have $\mathcal{O}(\sqrt[3]{g_{\mathcal{H}_{k+1}}})$ generators. From our induction hypothesis, we also know that $\mathcal{H}_k$ has $\mathcal{O}(g_{\mathcal{X}}^{3^k})$ generators. Thus, $\mathcal{H}_{k+1}$ has $\mathcal{O}((g_{\mathcal{X}}^{3^k})^3) = \mathcal{O}(g_{\mathcal{X}}^{3^k \cdot 3}) = \mathcal{O}(g_{\mathcal{X}}^{3^{(k+1)}})$ generators, which proves the statement. $\square$

**Theorem 1** (Enclosure of Large Language Models). *Given a large language model $\Phi$ and an input set $\mathcal{X}$ (14) with $g_{\mathcal{X}}$ generators, Alg. 1 computes an output set $\mathcal{Y}$ satisfying the problem statement stated in Sec. 2.5. The computational complexity is bounded by $\mathcal{O}\left(thd_V d_{\mathrm{model}} g_{\max} \kappa\right)$.*

*Proof.* The computation of $\mathcal{Y}$ is sound, as each step in Alg. 1 is outer-approximative (Lemma 1, Prop. 2, Prop. 3, (10), and (9)). The number of generators of all sets is bounded by $g_{\max}$. The overall computational complexity then follows from the largest matrix multiplication on the largest sets (10), which is given by aggregation of the attention heads given in line 6:

$$\underbrace{t \times hd_V}_{\text{Dimensions of concatenated } \mathcal{H}_{k,i}} \times \underbrace{hd_V \times d_{\mathrm{model}}}_{\text{Dimensios of aggregation matrix } W_\kappa^A} .$$

This matrix multiplication is applied to each generator (10), which are bounded by $g_{\max}$. As this bound can already be reached in any of the $\kappa$ transformer blocks, this computation is done at most $\kappa$ times. Thus. the final computational complexity is given by $\mathcal{O}(thd_V d_{\mathrm{model}} \cdot g_{\max} \cdot \kappa)$. $\square$

# D    ADDITIONAL EXPERIMENTS

In this section, we state further experiments and give all details about the dataset and our models. The medical safety dataset (Abercrombie & Rieser, 2022) is a small written English dataset consisting of risk-graded medical and non-medical queries that we split in $2,187$ training, $365$ validation, and $365$ test samples. We collapse the risk-levels into one class to enable binary text classification. The Yelp dataset (Zhang et al., 2016) consists of review texts, where we partition the dataset into $40,000/20,000$ training, $4,000/2,000$ validation, and $1,600/800$ testing samples, respectively. We train both datasets for binary classification referring to the labels as positive and negative. The positive label in the medical safety dataset refers to a valid medical query, while it refers to a positive review in the Yelp dataset. We use a BERT tokenizer (Devlin, 2018) to compute the mapping from the textual input to token identifier that we pass then to our model. Tab. 3 summarizes the used hyperparameters. All computations were performed in a docker container on an Intel® Core™ Gen. 11 i7-11800H CPU @2.30GHz with 64GB memory.

Table 3: Hyperparameters for considered models.

| Name | Variable | Model 1 | Model 2 | Model 3 | Model 4 |
|---|---|---|---|---|---|
| Dataset | - | Medical | Medical | Yelp | Yelp |
| Max. #tokens/sentence during verification | $t$ | 21 | 21 | 27 | 27 |
| Embedding dimension of model | $d_{\text{model}}$ | 8 | 8 | 8 | 8 |
| Number of transformer blocks | $\kappa$ | 2 | 1 | 2 | 1 |
| Number of attention heads | $h$ | 2 | 1 | 2 | 1 |
| Embedding dimension for query and key matrices | $d_{QK}$ | 4 | 8 | 4 | 4 |
| Embedding dimension for value matrix | $d_V$ | 4 | 8 | 4 | 4 |
| Maximum number of generators (order reduction) | $g_{\text{max}}$ | 1000 | 4000 | 1000 | 4000 |

Table 4: Comparison of verified embedding space with interval bound propagation (IBP), zonotopes (Z, Bonaert et al. (2021)), and polynomial zonotopes (PZ, ours).

| | Model 2 | | | Model 4 | | |
|---|---|---|---|---|---|---|
| | Verified Volume | | Time [s] | Verified Volume | | Time [s] |
| **Approach** | Mean | Max. | | Mean | Max. | |
| IBP | 0.00 | 0.00 | **0.01** | 0.00 | 0.00 | **0.01** |
| Z (baseline) | 1.00 | 1.00 | 0.76 | 1.00 | 1.00 | 0.54 |
| PZ ($\rho_{\text{lim}} = 2$) | 1.77 | 3.50 | 5.80 | 2.29 | 4.23 | 4.70 |
| PZ ($\rho_{\text{lim}} = 5$) | 2.16 | 3.83 | 11.16 | 3.51 | 9.61 | 9.26 |
| PZ ($\rho_{\text{lim}} = 10$) | 2.37 | 4.53 | 14.23 | 4.13 | 12.63 | 11.59 |
| PZ ($\rho_{\text{lim}} = 20$) | 2.66 | 5.35 | 18.67 | 5.35 | 20.34 | 14.53 |
| PZ ($\rho_{\text{lim}} = 50$) | 3.20 | 7.47 | 27.56 | 8.00 | 37.46 | 21.35 |
| PZ ($\rho_{\text{lim}} = 100$) | 4.09 | 11.33 | 36.94 | 11.62 | 64.37 | 29.50 |
| PZ ($\rho_{\text{lim}} = 200$) | 10.76 | 42.64 | 32.21 | 55.90 | 452.41 | 28.88 |
| PZ ($\rho_{\text{lim}} = 500$) | 18.60 | 89.44 | 31.13 | 123.62 | 1,406.54 | 26.36 |
| PZ ($\rho_{\text{lim}} = 1000$) | **28.79** | **158.75** | 31.16 | **137.41** | **1,606.65** | 26.83 |

Finally, we state the verification results on the second model of each dataset. For these models, we show the influence of the parameter $\rho_{\text{lim}}$ on the verified volume of the embedding space and the verification time in Tab. 4. As $\rho_{\text{lim}}$ increases, the verified volume increases as well at the cost of additional computation time. This can be done until the order reduction due to the maximum number of generators $g_{\text{max}}$ counteracts this advantage; thus, the outer-approximation induced by this order reduction becomes larger than the benefit of computing more precise enclosures. The parameter $g_{\text{max}}$ is determined by the memory constraints of the machine running the verification. We decided to show the mean and maximum verified volume instead of the standard deviation as this data is not normally distributed on this datasets with varying sentence lengths, and thus this statistic does not have a meaningful interpretation. Please note that setting $\rho_{\text{lim}} = 1$ corresponds the approach where no nonlinear dependencies are preserved as all higher-order terms are outer-approximated by an interval. Thus, we can always fall back to the approach by Bonaert et al. (2021) and our approach is thus at least as good as theirs.

