# OpenReview forum: "Towards Formally Verifying LLMs: Taming the Nonlinearity of the Transformer"
_ICLR.cc/2025/Conference — Submitted to ICLR 2025_

### Official Review · Reviewer_iaiX · 2024-11-03

**Soundness:** 3
**Presentation:** 2
**Contribution:** 3
**Rating:** 5
**Confidence:** 4

**Summary:**

The authors propose a novel transformer verification method based on matrix polyhedral zonotopes. Unlike the SOTA tool DeepT, which defines multiple zonotopes as the abstract domain, this work introduces the use of matrix polynomial zonotopes (a non-convex abstract domain) to encode the reachable set for each layer or function in transformers, achieving reduced approximation error. Additionally, to control the number of generators within the propagated zonotope abstract domain, the authors apply an existing order reduction method to enhance verification scalability with acceptable precision loss. Experimental results demonstrate its effectiveness and efficiency compared to DeepT.

**Strengths:**

- The problem (verification of Transformers) is important.
- The use of using non-convex abstract domain "matrix polynomial zonotope" to propagate the reachable set in Transformers is innovative and offers improved precision over existing SOTA methods.
- Experimental results support its effectiveness and efficiency.

**Weaknesses:**

- The paper lacks self-containment, particularly with the absence of a definition for the "enclose" function, which is utilized throughout the paper. However, I failed to find any information (definition or features) about it, even in the appendix.
- Without a description of the "enclose" function or its features, it is challenging to validate the soundness of the over-approximation via the enclose function, especially for the non-linear layers like softmax. While proofs are provided in the appendix, they lack clarity and fail to convincingly demonstrate soundness.
- A comparison with related methods would enhance the paper, particularly with approaches using different types of abstract domains and branch-and-bound methods (https://files.sri.inf.ethz.ch/wfvml23/papers/paper_24.pdf). Though designed for NN verification, these methods may also apply to the networks evaluated in this work, suggesting that such a comparison could be beneficial.
- While polynomial abstraction is applied for NN-like layers (LIN+ACT), the authors do not specify how transformations between the matrix polynomial zonotope and polyhedron are handled. Is this trivial or non-trivial in your setting? Providing more detail on this would improve clarity.

The presentation requires great improvement. However, I may change my score depending on the authors' responses.

**Questions:**

Please address the weakness raised in **Weaknesses**.

Other minor comments:
1. Please consider adding explicit multiplication symbols for clarity. For example, in (4), QK^T might be easily confused with d_{QK}. Additionally, in Theorem 1, "the computation complexity is bounded by ?" No multiplication makes it look like a typo (what are t, h, and all the other symbols?) Authors should clearly and explicitly define these symbols in the theorem to make it self-contained.
2. In (3),  should it be L^{LIN} in (3) instead of L^L?
3. Line 503: a citation is missing.
4. The experiment setting is not clear to me. How did you compute the verified volume in Table 2?

---

### Official Review · Reviewer_znn3 · 2024-11-04

**Soundness:** 3
**Presentation:** 3
**Contribution:** 2
**Rating:** 5
**Confidence:** 3

**Summary:**

The paper tackles the complex issue of verifying large language models (LLMs) against adversarial attacks. The authors propose a new approach based on non-convex set-based computing, specifically utilizing polynomial zonotopes, to preserve the nonlinear dependencies inherent in transformers. The framework allows for adjustable precision in verification through a single parameter, making it adaptable to varying computational resources and robustness requirements. This method aims to overcome the large approximation errors often seen with traditional convex relaxation techniques.

**Strengths:**

1. **Adaptive Precision Tuning**: The single-parameter tunability enables balancing verification precision with computational efficiency, which is highly practical for real-world applications.

2. **Scalability Demonstrated**: Experiments demonstrate that the approach scales well compared to zonotope and interval bound propagation methods, achieving significantly tighter enclosures and larger verified input spaces.

**Weaknesses:**

1. **Assumptions on Word Embeddings**: The approach assumes that word embeddings with similar meanings are sufficiently close in the embedding space. This may not universally hold, particularly in edge cases with nuanced or context-dependent synonyms.

2. **Single Focus on Adversarial Prompt Verification**: The paper primarily focuses on preventing adversarial prompts from bypassing safety checks, without exploring other safety risks such as jailbreaks, which also pose significant risks in LLMs. The set-based approach is limited to an input space near the unperturbed input.

3. **Limited Applicability to Large-Scale Models**: While promising, the current method is not feasible for the largest LLMs in practical use today, limiting the impact on widely deployed models.

**Questions:**

1. How can your method be adapted to verify robustness in contexts where word embeddings do not cluster neatly by meaning? Have you considered incorporating additional perturbation techniques beyond the simple $l_\infty$-ball around embeddings?

2. Could this approach be extended to detect and prevent other forms of unsafe behavior, such as jailbreaks, or would it require significant modifications?

---

### Official Review · Reviewer_5DGY · 2024-11-04

**Soundness:** 2
**Presentation:** 1
**Contribution:** 1
**Rating:** 3
**Confidence:** 3

**Summary:**

The paper aims to improve the formal verification for transformers by extending Matrix Polynomial Zonotopes to handle additional layers in the transformer architecture, compared to previous transformer verification methods using convex relaxation or zonotopes. A significant part of the story of this paper also involves the safety of large language models which, however, does  not look relevant to the methodological/empirical contribution of the paper.

**Strengths:**

* The paper extended Matrix Polynomial Zonotopes to handle additional layers in the transformer architecture.
* Empirical results showed improvement over zonotopes (achieving much larger verified volume on the embedding space given more running time), by using the proposed Polynomial Zonotopes for transformer verification.
* The story writing of the paper connected transformer verification to the safety of large language models, which might inspire future works.

**Weaknesses:**

* The paper is misleading with a story in the context of large language models. The problem actually solved by the paper (verification for the transformer architecture) is not new compared to existing works (verification for the transformer architecture in a classification setting with embedding-space perturbations). The proposed method is not applicable to real large language models and experiments do not involve any large language model. The authors claimed that “We evaluate our approach on four large language models”, but this is clearly a false claim, as these models are shallow Transformers (with no more than 2 Transformer blocks; see Table 3) trained from scratch by the authors on a tiny dataset (2187 training examples according to Appendix D). Such models are in no way “large language models”.
* Technical novelty is unclear. From what I see, the methodology contribution of the paper is on extending “Matrix Polynomial Zonotopes” from existing works to a few more operators (attention, softmax, layer norm) in the Transformer architecture. Given the current writing, I cannot tell any technical novelty, as it is unclear what’s challenging and what’s new in the new derivation. (See below for my comments regarding the writing.)
* Baselines are limited (only IBP, and zonotopes (Bonaert et al. (2021))). Experiments should include more recent works such as Wei et al., 2023; Shi et al., 2024; Zhang et al., 2024 which have already been mentioned in the paper but not empirically compared, as well as experiments on transformer models used in previous works. The paper is also missing citations for IBP for neural network verification (https://proceedings.mlr.press/v80/mirman18b/mirman18b.pdf, https://arxiv.org/abs/1810.12715)
* The background and method sections of the paper have minimal readability. The authors put many theoretical results or quoted some equations from existing works without any explanation. The paper should have a self-contained narrative in the main text for readers to understand the core ideas and contributions. The authors should not simply refer to proofs in the appendix.
  * In Definition 2 in Section 2, the authors quoted a very complicated definition from Ladner et al., 2024 without any explanation. The authors only showed some examples without formally explaining the terms in the long equation.
  * In Section 3, the authors showed some theoretical results for the derivation of exclosures for several operations in Transformers. The authors simply put the results there without any explanation (What do the enclosures look like? What’s challenging and what’s new in the derivation? ...).

**Questions:**

See above.

---

### Official Review · Reviewer_w7pC · 2024-11-04

**Soundness:** 1
**Presentation:** 1
**Contribution:** 2
**Rating:** 3
**Confidence:** 4

**Summary:**

The authors propose a new algorithm for verifying transformers. They use Matrix Polynomial Zonotopes, a generalization of the existing Zonotope domain that, unlike Zonotopes, allows for higher-order terms of generators or noise variables. The authors propose output enclosures for different constituent functions (such as softmax and multi-head attention) within the transformer architecture, which serve as the building blocks of the proposed verifier. The experiments suggest that the proposed Matrix Polynomial Zonotopes are more precise for verifying transformers than existing works that use either intervals or Zonotopes.

**Strengths:**

- The problem of verifying the safety of LLMs is important and critical for their real-world applications.
- The Matrix Polynomial Zonotopes are theoretically more expressive than the traditional Zonotope domain, and as expected, they outperform it in experiments.

**Weaknesses:**

**Weak experimental section**

Q1. The hyperparameter values in Table 3 seem small. For example, the 'embedding dimension for query and key matrices' appears to be a single digit. How do these values compare with those of real-world LLMs?

Q2. Even on this smaller network, the proposed algorithm with $\rho_{lim} = 2$ is 29.23x slower than the Zonotope method on Model1. Can the authors elaborate on the scalability of the proposed method?

Q3. It is not surprising that Matrix Polynomial Zonotopes are more precise than Zonotopes, as suggested by the experimental results. However, Zonotopes are used in neural network verification not for their precision, but because they are fast and easy to handle. The authors themselves mention (lines 365 - 367) that 'many of these higher-order terms contribute only a little to the overall set and can be outer-approximated by an interval without losing much precision.' Why isn’t this an argument for using Zonotopes, which lack higher-order terms, over Matrix Polynomial Zonotopes, given that they are faster and likely to scale better for larger DNNs?

**Expressivity vs computational efficacy**

Q1. Following up on the previous discussion, what is the computational complexity of the output enclosure algorithms? For example, consider the steps mentioned in Fig. 2.

Q2. The parameter $\rho_{lim}$ is a key factor in balancing expressivity and computational efficiency, yet it does not seem to be properly defined and explained.

**Changing the title to avoid overclaiming**


Although verifying transformers is an important step, I do not believe it is the primary challenge in verifying large LLMs with billions of parameters. Authors should provide evidence that their method **scales** to real-world LLMs. Otherwise, they should seriously consider removing "verifying LLMs" from the title, as it creates a misleading impression.

**Paper is not self-contained**

In the technical section, the paper relies heavily on results (line 191), approaches (line 257), or definitions from other works. Because these critical components are not explained in the main paper, it becomes extremely difficult to follow.

**Related works**

The works cited in this paper are outdated and have been surpassed by more recent research. For example,

**Abstract Domain/set-based computation:** The Zonotope has been surpassed by the DeepPoly [1] (misplaced in optimization-based verifiers). Subsequently, multi-neuron abstraction was introduced in [2], and more recently, the DiffPoly [3] was proposed for hyperproperty verification.

**Branch & Bound-Based Verifiers:** Current state-of-the-art verifiers for local robustness properties, such as GCP-CROWN [4] and MN-BaB [5], are not mentioned.

[1] “An Abstract Domain for Certifying Neural Networks”, POPL, 2019.\
[2] “PRIMA: General and Precise Neural Network Certification via Scalable Convex Hull Approximations”, POPL, 2022.\
[3] “Input-Relational Verification of Deep Neural Networks”, PLDI, 2024.\
[4] “General Cutting Planes for Bound-Propagation-Based Neural Network Verification”, NeurIPS, 2022.\
[5] “Complete Verification via Multi-Neuron Relaxation Guided Branch-and-Bound”, ICLR, 2022.

**Questions:**

Refer to the Weaknesses section

---

### Author Response · Authors · 2024-11-27

Dear reviewers,

Thank you very much for your valuable comments.
We will address your reviews in a future version of this paper.

---

### Meta-Review · Area_Chair_drkW · 2024-12-22

**Metareview:**

The paper studies the formal verification of Transformer-based architecture using Matrix Polynomial Zonotopes based techniques. Although the verification algorithm itself is valid (by extending polynomial zonotopes to the operators used in Transformers), many reviewers have concerns about the overclaimed contributions and the missing comparison to some baselines. The techniques in this paper cannot be practically used in LLMs due to limited scalability, although the title and introduction appear to claim LLM verification as its main contribution. The technical novel of the work is also unclear since polynomial zonotopes are not fundamentally new, and the baselines used in the experiment are insufficient. The paper is clearly below the bar of acceptance, but all reviewers have provided constructive feedback to help improve the paper for submission to another venue.

**Additional Comments On Reviewer Discussion:**

All reviewers reached the consensus that the paper needs more improvements and cannot be accepted in its current form. The authors do not provide a detailed response.

---

### Decision · Program_Chairs · 2025-01-22

Reject